# Programming Optimization in Implantable Cardiac Monitors to Reduce False-Positive Arrhythmia Alerts: A Call for Research

**DOI:** 10.3390/diagnostics12040994

**Published:** 2022-04-15

**Authors:** Fabrizio Guarracini, Martina Testolina, Daniele Giacopelli, Marta Martin, Francesco Triglione, Alessio Coser, Silvia Quintarelli, Roberto Bonmassari, Massimiliano Marini

**Affiliations:** 1Department of Cardiology, S. Chiara Hospital, 38122 Trento, Italy; marta.martin@apss.tn.it (M.M.); alessio.coser@apss.tn.it (A.C.); silvia.quintarelli@apss.tn.it (S.Q.); roberto.bonmassari@apss.tn.it (R.B.); massimiliano.marini@apss.tn.it (M.M.); 2ULSS 6 Euganea, Camposampiero, 35012 Padova, Italy; martina.test@icloud.com; 3Clinical Unit, Biotronik Italia, Vimodrone, 20090 Milano, Italy; daniele.giacopelli@biotronik.com (D.G.); francesco.triglione90@gmail.com (F.T.); 4Department of Cardiac, Thoracic, Vascular Sciences & Public Health, University of Padova, 35128 Padova, Italy

**Keywords:** cardiac arrhythmias, implantable cardiac monitor, implantable loop recorder, oversensing, remote monitoring, undersensing

## Abstract

No studies have investigated whether optimizing implantable cardiac monitors (ICM) programming can reduce false-positive (FP) alerts. We identified patients implanted with an ICM (BIOMONITOR III) who had more than 10 FP alerts in a 1-month retrospective period. Uniform adjustments of settings were performed based on the mechanism of FP triggers and assessed at 1 month. Eight patients (mean age 57.5 ± 23.2 years; 37% female) were enrolled. In 4 patients, FPs were caused by undersensing of low-amplitude premature ventricular contractions (PVCs). No further false bradycardia was observed with a more aggressive decay of the dynamic sensing threshold. Furthermore, false atrial fibrillation (AF) alerts decreased in 2 of 3 patients. Two patients had undersensing of R waves after high-amplitude PVCs; false bradycardia episodes disappeared or were significantly reduced by limiting the initial value of the sensing threshold. Finally, the presence of atrial ectopic activity or irregular sinus rhythm generated false alerts of AF in 2 patients that were reduced by increasing the R-R variability limit and the confirmation time. In conclusion, adjustments to nominal settings can reduce the number of FP episodes in ICM patients. More research is needed to provide practical recommendations and assess the value of extended ICM programmability.

## 1. Introduction

Implantable cardiac monitors (ICMs) provide continuous heart rhythm monitoring, and their use is steadily increasing in clinical practice. In combination with remote monitoring (RM), they can allow the diagnosis of cardiac arrhythmias and early medical interventions. A growing body of literature recognizes the clinical benefits of this strategy in different clinical scenarios, such as unexplained syncope or palpitations, cryptogenic stroke, and atrial fibrillation management [1,2,3,4,5,6].

On the other hand, since ICMs are designed to be highly sensitive to arrhythmias, they can generate an excessive number of recordings transmitted by RM daily and could require significant consumption of healthcare resources for their review [7]. These devices have several programmable settings. However, no studies have investigated whether ICM programming optimization could reduce the amount of false-positive (FP) recordings, and practical recommendations are lacking.

In this case series, we applied uniform adjustments to nominal settings in patients with ICM affected by a relevant number of false arrhythmia alerts to preliminary test this hypothesis.

## 2. Materials and Methods

Among all patients implanted with an ICM (BIOMONITOR III, Biotronik, Berlin, Germany) and followed with daily RM technology (Home Monitoring, Biotronik) in our institution, we included in the present analysis those with more than 10 FP alerts in a 1-month retrospective observational period. All patients gave their informed consent in writing to analyze the data provided by remote monitoring. The study was carried out in accordance with the Declaration of Helsinki, applicable local law, and the European Directive for data protection (General Data Protection Regulation).

The BIOMONITOR III is a leadless ICM composed of a solid housing part and a silicone flexible part designed to increase the length of the sensing dipole up to 7.7 cm. The housing of the device is 77 × 9 × 5 mm with a volume of 1.9 cm^3^ and a mass of 4 g.

ICM continuously monitors the heart rhythm. The subcutaneous electrocardiogram is recorded with a snapshot of 60 s in case of automatic arrhythmia detection. The following are possible detection types: atrial fibrillation (AF), high ventricular rate (HVR), asystole, bradycardia, and sudden rate drop.

The nominal settings for the detection parameters are (i) the variability of the R-R interval exceeding 12.5% (threshold) for 6 min (confirmation time) for AF, (ii) 16 beats exceeding 180 beats per minute (bpm) for HVR, (iii) mean heart rate below 40 bpm for 10 s for bradycardia, and (iv) pause lasting more than 3 s for asystole. The sudden rate drop algorithm is turned off by default. The device sensitivity can also be programmed using a parameter named ‘Sensing Consult’ that provides a range of preset configurations to adjust the accuracy of R-wave detection by modifying the decay of the dynamic sensing threshold. These parameters can be modified in a wide range of values by device interrogation (Table 1).

During the 1-month observational period of the study, the devices were set with standard programming and followed remotely. Briefly, data are retrieved daily from the device through a wireless receiver for long-distance telemetry (CardioMessenger Smart, Biotronik, Berlin, Germany). The receiver forwards the data to a unique service center by connecting to the GSM (Global System for Mobile Communication) network. The Service Center anonymously decodes, analyzes, and uploads data on a secure website, with a complete overview for the attending hospital staff. Remote daily transmissions include daily recordings of detected arrhythmias. Transmitted alerts were reviewed on all working days by a technician who has been deeply trained in cardiac implantable electronic devices [8]. An arrhythmia alert was classified as FP when the adjudicator did not agree with the ICM interpretation. The number, type, and reason of FPs were collected.

Optimization of device programming was performed in patients with a high incidence of FP (more than 10 episodes in the 1-month observational period). False arrhythmia alerts were analyzed to describe the specific sensing/detection problem and identify the most common mechanisms of FP triggers. The settings modifications, based on empirical previous experience and algorithm technical characteristics, were then made uniform to the detected problem.

Patients were prospectively followed for 1 month to compare the number of FP alerts before and after programming optimization.

## 3. Results

Our database of ICM patients followed with RM identified eight patients (mean age 57.5 ± 23.2 years; 37% female) with a high incidence of FPs. Indications for ICM insertion were unexplained syncope (*n* = 4), unexplained palpitations (*n* = 3), and cryptogenic stroke (*n* = 1). The characteristics of the patients are summarized in Table 2.

During the 1-month observational period, 526 FP recordings were identified, including 398 (75.7%) false bradycardia in 5 patients and 128 (24.3%) false AF in 5 patients. The median number of FP per patient was 38 [interquartile range, 28–84]. No false asystole or HVR alerts were observed.

Three different sensing or detection problems were identified. In four patients, false bradycardia and AF alerts were caused by the undersensing of large premature ventricular contractions (PVCs) but with low amplitude, as shown in Figure 1. We modified the parameter ‘Sensing Consult’ from ‘Standard’ to ‘small PVC Sense’ in these patients. Basically, the latter configuration consists of a more aggressive decay of the dynamic sensing threshold after a sensed event to avoid undersensing of a close R-wave complex with lower amplitude. The following month, no further false bradycardia was observed compared to the 327 transmitted before programming optimization. In addition, false AF decreased significantly in 2 patients from 39 to 10 episodes, while the number of FPs did not decrease in one patient (12 vs. 29) due to persistent undersensing of PVCs.

The second detection issue that we observed was the undersensing of sinus rhythm R waves after high-amplitude PVCs, as shown in Figure 2. This mechanism of FP generation was found in 2 patients. We modified the parameter ‘Sensing Consult’ from ‘Standard’ to ‘Sense after large PVCs’. This configuration does not modify the decay of the sensing threshold but limits the initial sensitivity threshold to 62.5% of 0.75 mV if the peak of the sensed event is higher, making more difficult the undersensing of R-wave complexes after events with high amplitude. This modification eliminated false bradycardia episodes in one patient (from 11 to 0) and dramatically reduced them in the other (from 60 to 6), as reported in Figure 2.

The last problem we observed was the presence of atrial ectopic activity or irregular sinus rhythm, causing intermittent variability of the R-R interval above the 12.5% threshold. An example is depicted in Figure 3. This mechanism generated a total of 77 false AF alerts in 2 patients. We increased the R-R variability limit from 12% to 15% and the confirmation time from 6 to 10 min to solve this detection problem. This modification eliminated false episodes of AF in one patient (from 35 to 0) and dramatically reduced them in the other (from 42 to 5), as reported in Figure 3.

## 4. Discussion

ICMs are widely recognized as effective and safe tools for detecting bradycardia or tachyarrhythmias in patients with unexplained recurrent syncope, cryptogenic stroke, palpitations, or presyncope [1,2,3,4,5,6]. Several studies with different ICMs models have assessed the diagnostic power of ICMs. In a study by Maggi et al. [9] in 58 patients with transient loss of consciousness, ICM allowed the detection of an arrhythmic cause of syncope in 15 (26%) patients. Furthermore, the absence of arrhythmia recordings during syncope recurrence in 18 patients led to excluding an arrhythmic cause. A more recent single-center study conducted among 154 consecutive patients confirmed that ICMs allow a diagnosis in nearly two-thirds of patients and to start or modify a medical therapy in 39%. In 17% of patients who suffered asymptomatic arrhythmias, RM transmissions led to therapeutic interventions 3.8 months before the next scheduled in-office evaluation [10]. Robust clinical evidence has also been produced on the effectiveness of ICMs in patients with cryptogenic stroke. According to the CRYSTAL-AF study [11], ICM-based ECG monitoring is superior to conventional follow-up to detect AF in patients with cryptogenic stroke. Different studies reported a sensitivity and specificity in detecting patients with AF ranging from 96% to 100% and 67% to 86%, respectively [12,13,14,15]. Sensitivity was lower when considering the detection of all episodes of AF, ranging from 88% to 95%, with positive predictive values of around 70%.

Despite the proven usefulness of ICMs in detecting arrhythmias, their subcutaneous position may cause frequent FP recordings. A recent study conducted 695 remote transmissions from 559 patients demonstrated an incidence of FP ranging from 46% to 86%, depending on the indication [5]. No false asystole alert was recorded in our small group of patients, while most alerts (75.7%) were false bradycardia. Previous studies reported a high incidence of false asystole and bradycardia events, ranging from 29% to 66% at 1 year [10,15]. The only available multicenter study analyzed a cohort of 1470 patients monitored during a 6-month period, 59.8% of 14,086 total alerts were erroneous, with particularly high FP rates for AF (74.2%) and asystole (76.8%) alerts [16]. This excessive number of transmitted recordings results in a high workload for hospital personnel to review episodes [17]. It should be noted that the impact of resources consumption can change according to the specific features of the RM system. Automatic and daily transmissions of the home monitoring system associated with the ability to transmit up to six arrhythmic recordings per day of the device could exacerbate the information received by the clinic staff. However, a recent study estimated that an ICM capable of transmitting only one episode per day would fail to transmit the most relevant recording in about half of messages with multiple arrhythmias [18]. In this scenario, programming optimization that decreases the number of FP episodes could also reduce the workload of healthcare personnel without increasing the risk of missing significant findings.

Adequate R-wave amplitude at the time of implantation and during follow-up is crucial for correctly identifying heart rhythm. R-wave undersensing is a well-recognized cause of inappropriate detection of asystole/bradycardia, with an incidence ranging from 20% to 69% at 2 years from implantation [15,16,19,20,21,22,23,24,25,26,27]. Some studies looked at the patient’s characteristics or implant that could affect R-wave detection [19,20,21,22,23,24,25,26,27,28]. A recent study by De Coster et al. with 135 patients showed that the occurrence of inadequate R wave sensing was not related to the characteristics of the patients, such as sex, age, and BMI [19]. In another previous study, the obese category and women appear to have lower mean R wave amplitude values [28]. The location of the ICM insertion does not appear to significantly influence the R-wave amplitude [19,20,21,22,23,24,25,26,27,28]. Therefore, it seems reasonable that the choice of ICM position should depend on the patient’s habitus. Recently, Bisignani et al. proposed axillary implantation with a 45° angle relative to the midaxillary line. This approach appears to be a valid alternative to standard prepectoral ICM insertion [23].

The use of ICM with a long sensing vector has been proposed to increase the R wave amplitude [22,23]. Indeed, there is a strong positive correlation between electrode distance and electrocardiographic signal amplitude [29]. The ICM (BIOMONITOR II) we implanted in our cohort has a flexible ‘antenna’ that allows the extension of the sensing vector. According to a recent study [22], this design allowed to obtain a mean value of R wave amplitude of 0.70 mV (range: 0.22–1.82 mV), which is higher than those obtained with a shorter sensing vector [15]. Long-sensing vector technology appears to be particularly indicated in obese patients to avoid R wave undersensing [28]. It may also explain the absence of false asystole episodes found in our study. We also observed that R wave undersensing was predominantly caused by excessive low-amplitude PVCs or by the undersensing of sinus rhythm R waves after high-amplitude PVCs. Newer ICMs have a dynamic sensing threshold automatically adjusted after each sensed R wave. Moreover, it is possible to change this sensing setting in the BIOMONITOR III device. There are no studies available demonstrating the usefulness of program optimization in ICMs to reduce the number of FPs. The ‘small PVC Sense’ configuration allowed us to reduce PVC undersensing using an aggressive decay of the dynamic sensing threshold after a sensed event. We performed this setting change in the first group of patients with a significant reduction of FP events.

In undersensing sinus rhythm R waves after high-amplitude PVCs, we changed from ‘Standard’ to ‘Sense after large PVCs’ sensing configuration. This configuration reduces the initial sensitivity threshold to 62.5% of 0.75 mV if the peak of the sensed event is higher, favoring the detection of the following complex of lower amplitude. This modification seemed to be extremely effective in eliminating, or at least significantly reducing, FP bradycardia episodes in our two patients.

Atrial or ventricular ectopy activity, oversensing of P or T waves, and R waves misdetection may cause FP tachycardia or AF alerts [30]. Current available ICMs allow changing sensitivity programming to reduce the risk of oversensing. In our study population, most of the AF FP was due to the presence of atrial ectopy or irregular sinus rhythm. We modified standard programming by increasing the R-R variability limit and prolonging the confirmation time. With these changes, false AF episodes were dramatically reduced.

With the limited number of patients, the present experience should be considered a proof of concept study to describe the feasibility of preventing many FP alerts by device programming optimization. However, reducing sensitivity to arrhythmias due to FP episodes may also affect detection reliability and potentially cause underdetection of relevant arrhythmias, as already shown in implantable and wearable cardioverter-defibrillators [31,32], but also in ICMs [19]. The lack of a concomitant method of ECG monitoring to confirm safety and the short follow-up after reprogramming (1 month) of our analysis did not allow us to conclude this aspect. Further investigations in larger populations should be performed to explore the benefits and risks of programming optimization.

## 5. Conclusions

In our small study population, programming optimization of ICMs significantly reduced the number of FP events with potential benefits on the healthcare resources consumption for their review. More research is needed to provide practical recommendations to physicians and assess the potential value of extended ICM programmability on the data review burden required for managing large cohorts of patients followed with RM technology.

## Figures and Tables

**Figure 1 diagnostics-12-00994-f001:**
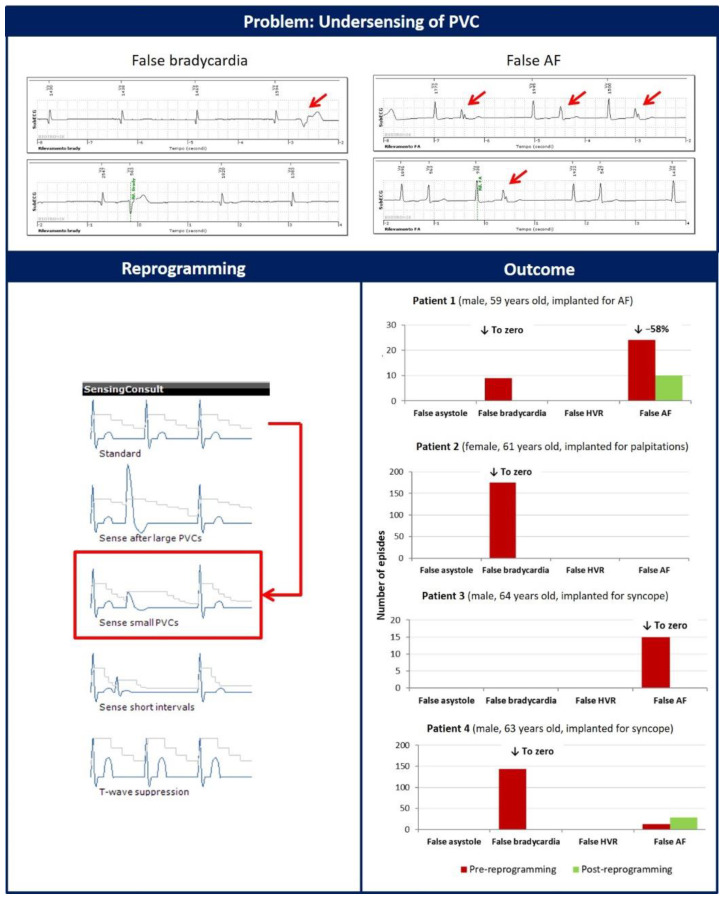
Red arrows indicate the undersensing of large, but low-amplitude premature ventricular contractions (PVCs). The reprogramming of the parameter ‘Sensing Consult’ from ‘Standard’ to ‘Sense small PVC’ significantly decreased the number of false bradycardia and atrial fibrillation episodes.

**Figure 2 diagnostics-12-00994-f002:**
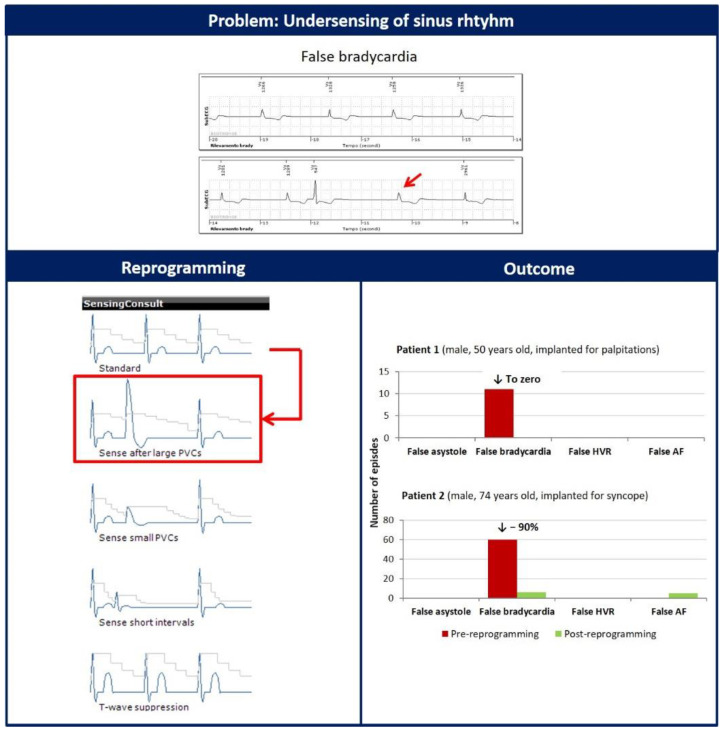
The red arrow indicates the undersensing of one sinus rhythm R wave following a high-amplitude PVC. Reprogramming of the parameter ‘Sensing Consult’ from ‘Standard’ to ‘Sense after large PVC’ dramatically decreased the number of false bradycardia episodes.

**Figure 3 diagnostics-12-00994-f003:**
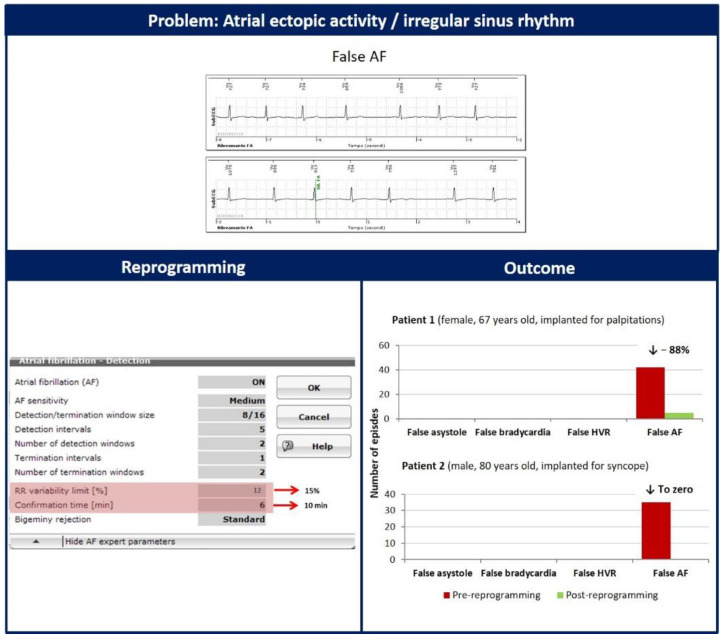
The presence of ectopic atrial activity or irregular sinus rhythm can cause false episodes of atrial fibrillation (AF). We increased the R-R variability limit from 12% to 15% and the confirmation time from 6 to 10 min to solve this detection issue.

**Table 1 diagnostics-12-00994-t001:** Parameters of the BIOMONITOR III detection algorithms with a range of available values and standard settings.

Parameter	Range of Values	Standard
Sensing Consult	Standard	Standard
	Sense after large PVCs	
	Sense small PVCs	
	Sense short intervals	
	T-wave suppression	
Atrial fibrillation (AF)	ON; OFF	ON
AF sensitivity	Low; Medium; High	Medium
RR variability limit	6; 9; 12; 15; 18%	12%
Confirmation time	1… (1)… 6; 10; 20; 30 min	6 min
Bigeminy rejection	OFF; Standard; Aggressive	Standard
High ventricular rate (HVR)	ON; OFF	ON
HVR limit	100… (10)… 200 bpm	180 bpm
HVR counter	8… (4)… 24; 32; 48 cycles	16 cycles
Bradycardia	ON; OFF	ON
Brady zone limit	30… (5)… 80 bpm	40 bpm
Brady duration	5… (5)… 30 s	10 s
Asystole duration	ON; OFF	ON
Asystole duration	2… (1)… 10 s	3 s
Sudden rate drop (SRD)	ON; OFF	OFF
SRD rate decrease	20… (10)… 70%	50%
SRD sensitivity	Low; Medium; High	Medium

Abbreviation: PVCs, premature ventricular contractions.

**Table 2 diagnostics-12-00994-t002:** Patient characteristics.

	Study Population (*n* = 8)
Age, years	57.5 ± 23.2
Female, *n* (%)	3 (37%)
ICM indication, *n* (%)	
Unexplained syncope	4 (50%)
Unexplained palpitations	3 (37%)
Cryptogenic stroke	1 (13%)
Comorbidities, *n* (%)	
Hypertension	5 (63%)
Coronary artery disease	1 (13%)
R wave amplitude, mV	0.41 ± 0.22

Abbreviation: PVCs, premature ventricular contractions.

## Data Availability

The data that support the findings of this study are available from the corresponding author, F.G., upon reasonable request.

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
