# Peer review of "Programming Optimization in Implantable Cardiac Monitors to Reduce False-Positive Arrhythmia Alerts: A Call for Research"

_diagnostics, 2022, doi:10.3390/diagnostics12040994_

Round 1
Reviewer 1 Report
The paper submitted for review is a very interesting one, assessing the subject of programming optimization in implantable cardiac monitors in order to reduce false-positive arrhythmia alerts.
Detection issues that may appear in patients wearing implantable cardiac monitors were correctly identified and accurately described.
However small, the study group was closely evaluated, with good results after reprogramming.
Thus, I find that this paper fulfils the criteria in order to be published.
Author Response
We thank the reviewer for his/her appreciation of our work.
Reviewer 2 Report
This article, which is a communication type, underlie that optimizing implantable cardiac monitors programming can reduce false-positive alerts. Appropriate programming of all implantable cardiac monitors is crucial in clinical practice. This small case series study is very interesting and very useful in clinical practice. The article is very well written. Discussions are in favors of research with this theme. It worth to be published.
Author Response
We thank the reviewer for his/her appreciation of our work
Reviewer 3 Report
The submission by Guarraccini, et al provides interesting results on a common shortcoming in remote management of implantable cardiac monitors since false alerts occur in as many as 46 % of implanted patients, as reported by the authors (ref 5 of their manuscript). Preventing such a large amount of false alerts is expected to spare time of physicians in charge of follow up, and also in focusing their attention to true harmful alerts.
Thus, regardless of the limited number of pts included , the results of the study, as well as the demonstrative exemples the authors included in the manuscript make the submission interesting for a broad readership.
Two points may improve the submission, I believe:
1 the authors should add more information on the partnership with Biotronik their study involved
2 A limitation of the study section should be added. Reprogramming detection of false alerts may impact reliability of arrhytmia detection and , in turn may result in potential underdetection of arrythmias. The authors should discuss this point, with , e.g, adding a few references on this topic:
- De Coster, et al, Pacing Clin Electrophysiol 2020; 43: 511
- - Kokubun et al, Circ J 2020; 85: 79
- - Lee et al, Cin Res Cardiol 2013; 102: 923
Reviewer 4 Report
The analysis concerning reprogramming implantable cardiac monitors (ICM) are needed. The trial confirming safety of ICM optimization should be performed. The best validation method seems to be implantation of 2 ICMs one with standard settings and the second constantly undergoing reprogramming adequately to the problems recognized in patients for reduction of false positive arrhythmias.
In the present study authors reduced false positive arrhythmias but did not include another tool (method of ecg monitoring) to confirm safety of reprogramming. I have questions concerning follow-up observations:
- were there any episodes of syncope, presyncope, palpitations, cryptogenic stroke after device optimization?
- were there significant tachy- or bradyarrhythmias identified in analyzed patients?
